# Peer review of "RETRACTED: A Soft Computing View for the Scientific Categorization of Vegetable Supply Chain Issues"

_logistics, 2022_

Round 1

Reviewer 1 Report

The paper presents interesting point of view related to scientific categorization of elements of supply chain connected with food industry (vegetables, exactly). The authors did a good job gathering and sorting information about this problem. It is well done study of literature related to the research topic.

I have some comments connected rather with the structure and language of this paper:

·   LANGUAGE. It is scientific paper, so for me, you should not use “we”, “our” in the text. In such documents, passive voice is recommended.

·   PURPOSE. It is not clearly presented in the text. It must be underlined.

· CONCLUSION. This chapter now is rather a summary. You present information what was done in the paper. It is the same, as purpose. Conclusion should be underlined in this text. You should compare the purpose and results of your work.

I have found 2 small problems in the text:

·  Page 2, line 51: “manner, No order”. You have a comma and capital letter. I think it should be changed into small letter.

·  Figure 7. The title – “chain issues. creation, handling,” I think it should be “:” after issues. Or maybe capital letter

Reviewer 2 Report

The topic is interesting, but this manuscript does not respond to the structure of a scientific article, nor does it conform to a systematic review.

1) Reduce the abstract, this should be a summary of the article, from the objective to the conclusions.

2) The introduction becomes diffuse, failing to make clear the objective of the work.

3) A review of the previous literature should be considered.

4) Above all, establish a method section, explaining how you will obtain your results or proposals.

5) The discussion should highlight the value of your work, explaining what this manuscript contributes in comparison to the background literature.

6) The conclusion should reinforce your results in terms of a clear objective.

7) See: https://www.mdpi.com/journal/logistics/instructions#preparation

Reviewer 3 Report

This study has suggested a novel approach and complete scientific classification of Vegetable Supply Chain concerns about Soft Computing. View for three delegate supply chains: Cruciferous vegetable, Dark green leafy vegetable, and Tomatoes. That assembled the scientific type in light of different parts to arrange Vegetable Supply Chain issues as per how they can be demonstrated utilizing Soft Computing methodologies. These parts are centered around recognizing the chain phase (creation, handling, dissemination, and retail) and the particular VSC issue to be tended to (e.g., vehicle steering issues in the appropriation phase). To check the strength of the scientific categorization, the authors classified VSC issues with SC techniques, particularly in the creation, handling, dispersion, and retail, scientific categorization. In outline, the scientific classification investigation proposes that there is no group of SC techniques that best suits all VSC issues. Even the authors express the requirement for a correlation system that permits the portrayal and examination of the exhibition of various SC strategies in different inventory network issues.

The work is well done but I have some remarks:

- The figures should be review, the dimension are variable

Moderate English changes are required

Round 2

Reviewer 2 Report

Dear authors, there are still unresolved weaknesses.

1) They raise an objective that is still unclear, but could be clarified if they manage to propose a more precise research methodology/method.

2) A research methodology/method remains absent, how do you achieve a scientific classification of VSC concerns about SC?

2.b) I have doubts whether we are dealing with an original article or a review article, and its methodological precision could support this clarification.

3) If you pose "The time was somewhere between 2012 and 2020, and the bibliographic assets looked at where the Web of Science, Scopus, and Google Scholar". how many documents are extracted and used in your analysis. This notorious lack of information does not allow you to clearly value your work.

4) The structure of the article is still quite unusual, many redundant aspects (sentences that are repeated over and over again). But little clarity of research methods or review, and value of the work with respect to other existing literature.

5) There is still no discussion, comparing this manuscript with other literature, and accounting for its value in terms of its contribution to science.
